# Evolutionary Patterns of the Chloroplast Genome in Vanilloid Orchids (Vanilloideae, Orchidaceae)

**DOI:** 10.3390/ijms24043808

**Published:** 2023-02-14

**Authors:** Young-Kee Kim, Se-Hwan Cheon, Ja-Ram Hong, Ki-Joong Kim

**Affiliations:** Division of Life Sciences, Korea University, Seoul 02841, Republic of Korea

**Keywords:** plastomes, SSC contraction, gene loss, gene relocation, Orchidaceae, Vanilloideae

## Abstract

The Vanilloideae (vanilloids) is one of five subfamilies of Orchidaceae and is composed of fourteen genera and approximately 245 species. In this study, the six new chloroplast genomes (plastomes) of vanilloids (two *Lecanorchis*, two *Pogonia*, and two *Vanilla* species) were decoded, and then the evolutionary patterns of plastomes were compared to all available vanilloid plastomes. *Pogonia japonica* has the longest plastome, with 158,200 bp in genome size. In contrast, *Lecanorchis japonica* has the shortest plastome with 70,498 bp in genome size. The vanilloid plastomes have regular quadripartite structures, but the small single copy (SSC) region was drastically reduced. Two different tribes of Vanilloideae (Pogonieae and Vanilleae) showed different levels of SSC reductions. In addition, various gene losses were observed among the vanilloid plastomes. The photosynthetic vanilloids (*Pogonia* and *Vanilla*) showed signs of stage 1 degradation and had lost most of their *ndh* genes. The other three species (one *Cyrotsia* and two *Lecanorchis*), however, had stage 3 or stage 4 degradation and had lost almost all the genes in their plastomes, except for some housekeeping genes. The Vanilloideae were located between the Apostasioideae and Cypripedioideae in the maximum likelihood tree. A total of ten rearrangements were found among ten Vanilloideae plastomes when compared to the basal Apostasioideae plastomes. The four sub-regions of the single copy (SC) region shifted into an inverted repeat (IR) region, and the other four sub-regions of the IR region shifted into the SC regions. Both the synonymous (dS) and nonsynonymous (dN) substitution rates of IR in-cooperated SC sub-regions were decelerated, while the substitution rates of SC in-cooperated IR sub-regions were accelerated. A total of 20 protein-coding genes remained in mycoheterotrophic vanilloids. Almost all these protein genes show accelerated base substitution rates compared to the photosynthetic vanilloids. Two of the twenty genes in the mycoheterotrophic species faced strong “relaxed selection” pressure (*p*-value < 0.05).

## 1. Introduction

One of the largest angiosperm families, Orchidaceae, consists of five subfamilies, 736 genera, and about 28,000 species [1]. The family is distributed worldwide and contains some mycoheterotrophs and many epiphytes. Some 43 genera among 736 genera contain mycoheterotrophs [2]. The mycoheterotrophic orchids are found in three subfamilies; Vanilloideae, Orchidoideae, and Epidendroideae. Vanilloideae comprises two tribes (Pogonieae and Vanilleae) [3]. Pogonieae contains five genera and 76 species, and Vanilleae consists of nine genera and 169 species [3]. Five genera (*Cyrtosia*, *Erythrorchis*, *Galeola*, *Lecanorchis*, and *Pseudovanilla*) of Vanilloideae are mycoheterotrophs [2]. Among the Vanilloideae, two species of *Pogonia*, one species of *Cyrtosia*, and two species of *Lecanorchis* are distributed in South Korea [4].

*Pogonia*, a genus of Pogonieae, consists of five photosynthetic species [3], which are distributed in the temperate regions of East Asia and eastern North America [5]. *P*. *japonica* and *P*. *minor* are native to the sunny and wet areas of southern Korea [4]. *Lecanorchis,* a genus of Vanilleae, comprises 20 obligate mycoheterotrophic species [2,3], which are distributed in Southeast Asia, the Pacific Islands, China, Japan, and South Korea [5,6]. *L*. *kiusiana* and *L*. *japonica* are native to South Korea [4]. *Vanilla*, which is the largest genus in Vanilloideae, contains 105 species, which are distributed in Southeast Asia, South America, and Africa [3,7]. *Vanilla* species are diversified into climbing vines, and some species are cultivated in the tropics as fragrant fruit crops [8].

A large number of Orchidaceae plastomes have been reported since the first report of the *Phalaenopsis aphrodite* plastome [9]. However, most of the reported orchid plastomes are confined to some genera containing mycoheterotrophs (*Corallorhiza* and *Neottia*) or horticulturally cultivated species (*Cymbidium*, *Dendrobium*, and *Phalaenopsis*) [10,11,12,13,14,15,16,17]. Orchid plastomes show a wide range of gene contents. For example, *Cattleya* (Epidendreae), *Cephalanthera* (Neottieae), *Cymbidium* (Cymbidieae), *Kuhlhasseltia* (Cranichideae), *Neofinetia* (Vandeae), and *Vanilla* (Vanilleae) plastomes do not contain various *ndh* genes [18,19,20,21,22]. The mycoheterotrophic orchid plastomes that are founded in *Corallorhiza* (Epidendreae), *Cyrtosia* (Vanilleae), *Epipogium* (Nervilieae), *Gastrodia* (Gastrodieae), *Neottia* (Neottieae), and *Rhizanthella* (Diurideae) are smaller in size and have fewer genes than photosynthetic orchids [10,11,15,23,24,25,26,27].

Thus far, only four complete plastome sequences of vanilloid orchids have been reported. Three *Vanilla* species (*V*. *pompona*, *V*. *planifolia*, and *V*. *aphylla*) have been sequenced, and their plastomes do not have some of the *ndh* genes [22,28]. Moreover, a small single copy region (SSC) in three *Vanilla* plastomes was largely reduced, and similar reductions have also been reported in *Paphiopedilum* (Cypripedioideae) and *Hetaeria* (Orchidoideae) [16,20,29]. The only sequenced mycoheterotrophic vanilloid, *Cyrtosia septentrionalis,* experienced a plastome size reduction and gene losses similar to other mycoheterotrophic orchids of other subfamilies [25]. The *C*. *septentrionalis* plastome also shows several gene rearrangements or gene relocations [25]. In order to understand the phylogeny of orchids, complete plastomes were used to construct a phylogenetic tree [20,30,31]. However, a few genes (*psa*B, *rbc*L, or *psb*C) were used to construct a phylogenetic tree of vanilloids [32,33,34,35] because of the limitation of whole plastome sequences. A few nuclear regions, such as the internal transcribed spacer (ITS) and *xdh*, were used to search Vanilloideae’s phylogenetic relationship [33,36,37,38]. However, the phylogenetic study of Vanilloideae has not been studied intensively because of the limited taxon sampling.

In this study, first, we documented six new vanilloid plastomes: two photosynthetic *Pogonia* (*P*. *minor* and *P*. *japonica*), two mycoheterotrophic *Lecanorchis* (*L*. *kiusiana* and *L*. *japonica*), and two photosynthetic *Vanilla* (*V*. *madagascariensis* and *V*. *planifolia*). Second, we validated the modifications of vanilloid plastomes and compared them to other orchid plastomes, such as SSC reductions, gene relocations, and gene reductions. Third, we used the elucidated plastome sequences to construct a phylogenetic tree so that the phylogenetic position and the divergence date of major vanilloids could be discussed. Lastly, in order to elucidate the evolutionary modes of vanilloid plastomes, we compared the synonymous (dS) and the nonsynonymous (dN) base substitution rates between the relocated and un-relocated regions both in the photosynthetic species and in the mycoheterotrophic species. In addition, we evaluated differential selection pressures on the different regions and groups.

## 2. Results

### 2.1. General Features of Vanilloideae plastomes

The average coverage depths, voucher specimens, DNA numbers, and lengths of assembled plastomes are given in Table 1, and the NGS results are given in Appendix A. The coverage depth of newly sequenced species ranged from 76.9× (for *Lecanorchis japonica*) to 4394× (for *Vanilla madagascariensis)*. The plastome size of *L. japonica* was the shortest, with 70,498 bp, and the size of *Pogonia japonica* was the longest, with 158,200 bp. Almost all plastomes have a quadripartite structure, which is composed of an large single copy (LSC), an SSC, and two IRs’. The size variations of the plastome regions of ten Vanilloideae and four Apostasioideae species are graphically presented in Figure 1. One noticeable size variation is the extensive reduction of the SSC region in the photosynthetic vanilloids. The SSC region ranged from 1254 bp in *V. madagascarensis* to 5885 bp in *P. japonica*. The annotated plastome maps of seven vanilloids belonging to the genera of *Cyrtosia*, *Lecanorchis*, *Pogonia*, and *Vanilla* are shown in Figure 2.

Almost all the genes from the nonphotosynthetic vanilloid species were found to be pseudogenized or lost. For example, all photosynthesis-related genes were lost from the plastomes of two mycoheterotrophic *Lecanorchis* species. Only 32 housekeeping genes (*acc*D, *clp*P, *inf*A, *mat*K, *rpl*2, *rpl*14, *rpl*16, *rpl*20, *rpl*36, *rps*2, *rps*3, *rps*4, *rps*7, *rps*8, *rps*11, *rps*12, *rps*14, *rps*18, *rps*19, 16S rRNA, 23S rRNA, 4.5S rRNA, 5S rRNA, *trn*C-GCA, *trn*D-GUC, *trn*F-GAA, *trn*fM-CAU, *trn*I-CAU, *trn*N-GUU, *trn*Q-UUG, *ycf*1, and *ycf*2) are present in *L*. *japonica*. In addition to the 32 genes, the *trn*E-UUC gene is present in the *L*. *kiusiana* plastome. The plastome of heterotrophic *Cyrtosia* showed intermediate gene contents between *Vanilla* and *Lecanorchis*. A few photosynthesis-related genes or pseudogenes still remain in the *Cyrtosia* plastome, while all photosynthesis-related genes have been lost in *Lecanorchis*. The detailed gene contents of vanilloids and the other related orchids are given in Appendix A.

### 2.2. Comparative and Phylogenetic Analyses

In order to validate the structural changes in the vanilloid plastome, the SC-IR junctions were analyzed among 14 Vanilloideae and Apostasioideae species using an IRScope. In the photosynthetic species, the LSC-IRa and LSC-IRb junctions are located within the *rpl*22 gene and the intergenic spacer between *trn*H and *psb*A, respectively. In the nonphotosynthetic species, the LSC-IRa junction is located within *rps*19 (*Cyrtosia*) or *rpl*2 (*Lecanorchis*), and the LSC-IRb junctions are located near the *mat*K gene (Appendix A). In the typical orchid plastome, the SSC-IRa and SSC-IRb junctions are located within the *ndh*F gene and the *ycf*1 gene, respectively. The SSC-IRa/b junctions of the four Apostasioideae species follow the same typical orchid form. Both the SSC-IRa and the SSC-IRb junctions in *Lecanorchis* are also located within *ycf*1. However, these junctions occurred within *ccs*A (*Vanilla*) and *ndh*A (*Pogonia*) in photosynthetic vanilloids. In the *Cyrtosia* plastome, the SSC-IRa and the SSC-IRb junctions are located near *rrn*16 and *ndh*B, respectively.

The gene orders of 14 plastomes were analyzed using progressiveMauve. A total of ten gene rearrangement events were found (Table 2). The longest inversion event occurred between *trn*S-GCU and *ycf*3 in the LSC region, with a length of 58.591 kb. The shortest rearrangement occurred between *ycf3* and *trnS-GGA*, and the inverted length was 3.282 kb. In addition to the 10 gene rearrangement events, several gene relocations occurred in mycoheterotrophic vanilloids. For example, the 16S rRNA, 23S rRNA, 4.5S rRNA, and 5S rRNA genes are usually positioned in the IR region. However, those four genes were in the SSC region in *L. kiusiana* and *L*. *japonica*.

The phylogenetic relationship of the vanilloid species was identified using a maximum likelihood (ML) tree and a Bayesian inference tree, which were constructed using 83 genes (Figure 3 and Appendix A). Both trees support the relationship of Apostasioideae (Vanilloideae(Cypripedioideae(Orchidoideae,Epidendroideae))). Almost all nodes were supported by a 100% bootstrap value and a 1.0 Bayesian posterior probability. In the subfamily Vanilloideae, two monophyletic groups, corresponding to the tribe Pogonieae and the tribe Vanilleae, were recognized. In the tribe Vanilleae, two clades corresponding to the photosynthetic group (*Vanilla*) and mycoheterotrophic group (*Cyrtosia* and *Lecanorchis*) were further identified. The *Vanilla* clade was further divided into two subclades corresponding to leafless species (*V*. *aphylla* and *V*. *madagascariensis*) and leafy species (*V*. *pompona* and *V*. *planitifolia*). The branch lengths leading to the mycoheterotrophic vanilloids are relatively longer than the branch lengths leading to the photosynthetic vanilloids in the ML tree.

### 2.3. Divergence Time of Vanilloideae

Orchidaceae was inferred to have diverged from other Asparagales at 101.1 (85.8–117.2) mya (Appendix A). A basal subfamily, Apostasioideae, was estimated to have diverged from other subfamilies at 84.49 (64.5–101.8) mya. Then, the Vanilloideae was inferred to have diverged from other subfamilies at 76.09 (57.6–94.0) mya. The two tribes of Vanilloideae, Vanilleae, and Pogonieae were estimated to have separated from each other at 61.54 (45.1–80.0) mya.

### 2.4. Evolutionary Rate Comparison

The SSC contraction, mycoheterotrophic life cycle, and rearranged plastome regions are notable features in Vanilloideae plastomes because they are not found in other typical orchid plastomes. The regions that survived in a plastome after a mycoheterotrophic event of *Cyrtosia* and *Lecanorchis* are listed. The rearranged regions are also listed. These listed regions were tested to calculate the nonsynonymous/synonymous substitution rates, and the results are listed (Figure 4, Figure 5, Appendix A).

The genetic regions that were relocated from the typical Orchidaceae plastome to the Vanilloideae plastome are called rearranged regions in this study. There are eight rearranged regions: *ccs*A, *rpl*2, *rpl*22, *rps*7, *rps*12-3′, *rps*15, *rps*19, and *ycf*1. Among the eight regions, four (*rpl*2, *rps*7, *rps*12-3′, and *rps*19) were moved from the IR region to the SC region. The other four regions (*ccs*A, *rpl*22, *rps*15, and *ycf*1) were moved from the SC region to the IR region. In the case of nonsynonymous substitution rates (dN), the difference between the dN values of the rearranged group and the typical group does not seem related to the region shift. Meanwhile, the synonymous substitution rates (dS) value of the rearranged group differed from the typical group for regions that were shifted to IR. The IR-shifted group showed a much lower dS value than the typical group (Figure 4). A log-likelihood test (LRT) was performed between the null model and the alternative model. The *ycf*1 region was the only statistically significant region among the eight shifted regions (*p*-value < 0.05).

There are twenty conserved regions in the Vanilloideae mycoheterotrophic plastome: *clp*P, *inf*A, *mat*K, *rpl*2, *rpl*14, *rpl*16, *rpl*20, *rpl*36, *rps*2, *rps*3, *rps*4, *rps*7, *rps*8, *rps*11, *rps*14, *rps*18, *rps*19, *rps*12-3′, *rps*12-5′, and *ycf*2 (Figure 5). For all those regions, the dN and dS values were calculated using a PAML. As a result, the dN and dS values of nineteen regions, except for the dS value of *clp*P, were higher than regions from photosynthetic plastome. LRT was also performed between the null model and the alternative model so that five regions (*clp*P, *rpl*16, *rps*7, *rps*12-5′, and *ycf*2) were statistically significant regions among twenty conserved regions (*p*-value < 0.05).

RELAX was performed using twenty-eight regions to check which evolutionary pressure was applied to the plastome. As a result, a total of six regions (*clp*P and *ycf*2 of the mycoheterotrophic conserved regions; *rpl*2, *rps*12-5′, *rps*15, and *ycf*2 of the rearranged region) were statistically significant (*p*-value < 0.05). Only the *rpl*2 region showed an intensification selection, while the other five regions showed a relaxation selection among statistically significant regions (Table 3).

## 3. Discussion

### 3.1. Plastome Structure Evolution

The SSC contraction is an interesting feature of the Vanilloideae plastome (Figure 1, Figure 2). Though an SSC in a typical Orchidaceae is 10–18 kb in length, it ranges from 1.2–17 kb in length in the Vanilloideae plastome. However, the whole length of the Vanilloideae plastome is not much different from the typical Orchidaceae (Appendix A). The SSC contraction occurs mostly in the photosynthetic Vanilloideae. *Vanilla madagascariensis*, a photosynthetic member of Vanilleae, possesses the shortest SSC, which is 1.2 kb in length. The longest SSC of the photosynthetic Vanilloideae is 5.3 kb in *Pogonia japonica*. The SSC contraction event in photosynthetic Vanilloideae occurs in two tribes (Pogonieae and Vanilleae). Interestingly, the two tribes experience different levels of SSC contraction. In Pogonieae, *Pogonia*’s SSC is 5.3 kb in length. However, the length of *Vanilla*’s SSC ranges from 1.2–2.1 kb. *Pogonia* is treated as a basal taxon in the maximum likelihood and Bayesian phylogenetic trees. Because the SSC of *Vanilla* is shorter than *Pogonia*’s, SSC contraction may be more extreme in the crown taxa of Vanilloideae.

The SSC contractions are common events in Vanilloideae. However, the SSC contraction in photosynthetic orchids is a rare evolutionary event from three other orchid subfamilies. It has only been reported in three species of *Paphiopedilum* (Cypripedioideae) and one species of *Hetaeria* (Orchidoideae) [16,20,29]. *Paphiopedilum*’s SSC ranges from 2.4–5.1 kb, and *Hetaeria*’s SSC is 2.3 kb. In photosynthetic orchids, SSC contraction is found in only three genera (*Paphiopedilum*, *Pogonia*, and *Vanilla*). Though very few cases are observed in photosynthetic orchids, most mycoheterotrophic orchids, such as *Neottia* (*N*. *acuminata*, *N*. *camtschatea*, *N*. *listeroides*, and *N*. *nidus*-*avis*), *Epipogium* (*E*. *aphyllum*, and *E*. *roseum*) have plastomes with shortened SSCs [11,15,26]. However, SSC contraction in mycoheterotrophs is an additional phenomenon that comes after gene reduction. Moreover, mycoheterotrophic Vanilloideae (*Cyrtosia*, *Lecanorchis*) has an SSC of 10–14 kb in length [25]. The SSC contraction event might be an independent event in Vanilloideae.

Another important event of the Vanilloideae plastome is gene reduction. In the case of Vanilloideae plastome, *ndh* genes are deleted or pseudogenized like in other Orchidaceae genera, such as *Cymbidium*, *Goodyera,* and *Neofinetia* [17,21,22,29]. Except for the *ccs*A gene of *Vanilla madagascariensis* (which is affected by SSC contraction), gene components of photosynthetic Vanilloideae (*Pogonia* and *Vanilla*) are similar to those of other photosynthetic orchids, such as *Habenaria*, *Phalaenopsis* [9,39].

On the other hand, mycoheterotrophic Vanilloideae (*Cyrtosia* and *Lecanorchis*) have fewer plastome genes than photosynthetic Vanilloideae. Mycoheterotrophs have experienced plastome gene reduction, which can be categorized by stages of degradation [40,41,42,43]. Orchids that lack the *ndh* gene can be categorized as degradation stage 1. In degradation stage 2, photosynthesis-related genes (*pet*, *psa*, *psb*, *rpo*) are usually reduced or pseudogenized [40,43]. The mycoheterotrophic Vanilloideae, *Cyrtosia septentrionalis*, was treated as an intermediate stage between stage 2 and stage 3 reduction [25]. In stage 3, only housekeeping genes, such as *trn*, *rps*, and *rpl*, remain in the plastome. A range of 27–33 genes remains in the orchid plastomes at degradation stage 3. It can be found in *Epipogium*, *Gastrodia*, and *Rhizanthella* [24,26,27]. *Lecanorchis japonica* and *L*. *kiusiana* have 32 or 33 genes, respectively, indicating that *Lecanorchis* has experienced stage 3 degradation.

Furthermore, rRNA gene relocation from the IR to the SSC region occurred in mycoheterotrophic Vanilloideae (Figure 2). Four rRNAs (4.5s rRNA, 5s rRNA, 16s rRNA, and 23s rRNA) usually stayed in the IR of a typical angiosperm plastome [9,44,45]. However, the relocation of eight genes (4.5s rRNA, 5s rRNA, 16s rRNA, 23s rRNA, *rps*7, *rps*12, *rps*15, and *ycf*1) was observed in *Cyrtosia septentrionalis* [25]. A similar relocation was discovered in the *Lecanorchis kiusiana* and *L*. *japonica* (Figure 2). In the *Lecanorchis* plastome, *rps*7 and *rps*12 were not moved to the SSC region, counter to the *Cyrtosia* case. The *rps*15 gene was deleted in the *Lecanorchis* plastome. This type of rRNA relocation in Orchidaceae has only been observed in *Rhizanthella gardneri* [24]. Four rRNAs, *rps*7, *rps*12, and *ycf*1, were moved from the IR to the SSC region in *Rhizanthella*. Other mycoheterotrophic orchids, such as *Corallorhiza*, *Epipogium*, *Gastrodia*, or *Neottia*, did not experience this rRNA relocation [10,11,15,23,26,27].

Genome rearrangement is another notable feature of the Vanilloideae plastome. In order to document the structural rearrangement, the Vanilloideae and Apostasioideae plastomes were compared. A total of ten rearrangements were found (Table 2) among species. The ten rearrangements have been plotted in the phylogenetic tree (Figure 6). In the genus *Neuwiedia* of Apostasioideae, the plastome structure is identical to other typical orchid plastomes. No structural rearrangements or gene reduction occurred in this taxon. However, *Apostasia*, the other genus of Apostasioideae, shares two rearrangements with *Pogonia* and *Cyrtosia* of Vanilloideae: rearrangement A (*ycf*3-*trn*S-GGA, 3.282 kb) and rearrangement B (*trn*S-GCU-*ycf*3, 58.591 kb). One more independent rearrangement (rearrangement D-*acc*D, 6.658 kb) is detected in *Apostasia*.

*Pogonia* and *Vanilla* share three rearrangements: G (*rpl*32-*ndh*F, 4.626 kb), I (*ycf*1-*ndh*A, 6.469 kb), and J (*ycf*1, 7.284 kb). *Cyrtosia* shares rearrangement I and J with *Pogonia* and *Vanilla*. *Lecanorchis* shares rearrangement J only with other Vanilloideae. *Pogonia* independently has rearrangement H (*trn*L-UAG-*ndh*A, 8.899 kb). This different rearrangement sharing may be caused by variations in SSC length. The *Pogonia* plastome has a longer SSC than *Vanilla*. In it, the *ndh*A region is generally located in the SSC region. *Pogonia*’s SSC region is long enough to preserve the *ndh*A region, but *Vanilla*’s SSC region is not. Thus, rearrangement H, which includes the *ndh*A region, occurs in the genus *Pogonia* only. In the case of mycoheterotrophic species, *Cyrtosia* has three independent rearrangements: rearrangements C (*trn*S-GCU-*rbc*L, 24.647 kb), E (*ndh*B-*rps*12, 4.626 kb), and F (*trn*V-GAC, 4.037 kb)). *Lecanorchis* does not show any other independent rearrangement due to its extremely degraded plastome. Thus, it seems likely that *Lecanorchis* had the common Vanilloideae rearrangements (such as G, H, I, and J) first and later experienced rearrangement losses due to the gene losses (Figure 6). However, it is difficult to argue which events occurred first due to the limited plastome sequence availability.

### 3.2. Evolutionary Rates and Selection Pressure

Eight relocated regions were affected by IR expansion/contraction (the *ccs*A, *rpl*22, *rps*15, and *ycf*1 regions moved to IR; the *rpl*2, *rps*7, *rps*12-3′, and *rps*19 regions moved to SC). The substitution rates of those eight regions were compared between the reference taxa (typical position of regions) and test groups (relocated). The nonsynonymous substitution rates (dN) are not different between the reference and test groups. However, the synonymous substitution rates (dS) are much lower in the test group than in the reference group in the case of the regions that were moved to the IR region. Similarly, the other four regions of the test group, which were moved to the SC region, show higher dS values than the reference group (Figure 4).

In the general angiosperm plastome, the IR strengthens the stability of the plastome structure. This structural stability is commonly supported by the copy-dependent repair mechanism (flip-flop recombination) stability [46,47]. For that reason, regions in IR show lower substitution rates or genetic diversity. Previous studies of other angiosperms, such as *Sesamum* (Lamiaceae) and *Pelargonium* (Polygonaceae), support the lower substitution rates of IR regions [48,49].

Mycoheterotrophic species usually have decreased plastome size and fewer plastome genes. Previous studies categorized mycoheterotrophic species by the level of plastome degradation [40,43]. According to those criteria, the *Lecanorchis* plastome can be categorized as stage 4 degradation, also known as the “stationary” stage [43]. In twenty retaining gene regions of *Lecanorchis*, the dN and dS substitution rates were compared between the test (mycoheterotrophic taxa; *Cyrtosia* and *Lecanorchis*) and reference groups (other photosynthetic relatives). Nineteen regions, except for *clp*P, have much higher substitution rates. There was a similar result in the genus *Rhizanthella*, which is an obligate mycoheterotroph of Ochidoideae [24]. In the case of the mycoheterotrophic *Corallorhiza*, however, it is reported that the w-ratio is not different from its photosynthetic relatives [50]. Although almost every region (except the only *clp*P region) shows elevated substitution rates, there are just five statistically significant regions in both the dS and dN comparisons (*clp*P, *rpl*16, *rps*7, *rps*12-5′, and *ycf*2). Therefore, this is not enough evidence to claim that the mycoheterotrophic life cycle affects retention regions’ elevated substitution rates.

The elevated substitution rates may be affected by selection pressure. In mycoheterotrophs, two regions (*clp*P and *ycf*2) show significant *p*-values (*p*-value < 0.05), which were acquired by RELAX. Those two regions were under relaxation selection pressure (K < 1), which means that the surviving regions in mycoheterotrophic species’ plastomes tend to be removed. Four regions under the position effect (*rpl*2, *rps*15, *rps*12-3′, and *ycf*1) showed significant p-values. Two regions (*rps*15 and *ycf*1), which were moved to the IR region from the SC region, experienced relaxation selection (K < 1), and the substitution rates were decreased in both. The other two regions (*rpl*2 and *rps*12-3′) show accelerated substitution rates but different selection pressure: *rpl*2 experienced intensification selection pressure, and *rps*12-3′ experienced relaxation selection pressure.

### 3.3. Phylogenetic Position and Time Estimation

In the maximum likelihood and Bayesian inference phylogenetic trees, *Epipogium*, *Gastrodia*, and *Rhizanthella*, which are mycoheterotrophs, have more elongated branch lengths compared to other photosynthetic taxa (Figure 3). Though the above three genera have extremely degraded plastomes, they have much fewer informative sites of the data matrix than other taxa. This leads to an elongated branch in the phylogenetic tree. However, mixed genera, such as *Corallorhiza* or *Neottia*, which contain both the photosynthetic and mycoheterotrophic species, have similar branch lengths to other typical clades, which are not significantly different. Because Vanilloideae also contains photosynthetic and mycoheterotrophic species, we also expected a similar branch length. However, the actual branch length of Vanilloideae seems to be elongated, especially in the genera with extremely degraded plastomes (*Cyrtosia* and *Lecanorchis*). Moreover, the branch length is longer in Vanilleae than in Pogonieae. This is presumed to be because the mycoheterotrophic genera are included in Vanilleae.

The whole chloroplast genome was used to estimate the derived time of Orchidaceae [20,30,51]. Vanilloideae was derived at 76.09 (57.6–94.0) mya (Appendix A) in this study. Though this estimation is much more recent than some from other previous studies, it is within the error range [20,30]. *Cyrtosia* and *Lecanorchis* were derived at 32.75 (18.49–52.45) mya. These two mycoheterotrophic genera have their plastomes at degradation stages 3 to 4. Other genera with plastomes at degradation stage 3 or 4, such as *Gastrodia* (Gastrodieae), *Epipogium* (Nervileae), and *Rhizanthella* (Diurideae), were derived at 39.17 (29.10–50.54) mya, 39.17 (29.10–50.54) mya, and 36.52 (25.45–54.49) mya, relatively. These estimations are very similar to the estimated time of mycoheterotrophic Vanilloideae (*Cyrtosia* and *Lecanorchis*). Old clades tend to have severe plastome degradation since they have had enough time to accumulate mutations [43].

Other mycoheterotrophic taxa, which have only slightly degraded plastomes, were derived much more recently than taxa with stage 3 or 4 degradations. *Corallorhiza* was derived at 21.45 (12.7–30.29) mya, *Aphyllorchis* was derived at 31.62 (14.15–52.01) mya, *Neottia* was derived at 25.66 (15.40–38.76) mya, and *Cymbidium* was derived at 25.36 (15.03–36.61) mya. *Aphyllorchis* (Neottieae), *Cymbidium* (Cymbidieae), *Corallorhiza* (Epidendreae), and *Neottia* (Neottieae) are composed of obligate mycoheterotrophs, mixotrophs, and even green relatives. Except for *Aphyllorchis*, the other three genera were derived slightly recently, before 30 mya. This result supports previous studies that claim that mycoheterotrophic orchids were derived recently, except for those with stage 3 or 4 degradations [20].

## 4. Conclusions

We reported six new chloroplast genomes (plastomes) of vanilloids (two *Lecanorchis*, two *Pogonia*, and two *Vanilla* species). Among these plastomes, *Pogonia japonica* has the longest plastome, with 158,200 bp in genome size. In contrast, *Lecanorchis japonica* has the shortest plastome, with 70,498 bp in genome size. The vanilloid plastomes have a typical quadripartite structure, but the small single copy (SSC) region was drastically reduced. The two different tribes of Vanilloideae (Pogonieae and Vanilleae) showed different levels of SSC reduction. In addition, various gene losses were observed among the vanilloid plastomes. The photosynthetic vanilloids (*Pogonia* and *Vanilla*) showed signs of stage 1 degradation and had lost most of their *ndh* genes. However, the other three species (one *Cyrtosia* and two *Lecanorchis* species) had stage 3 or stage 4 degradation and had lost almost all the genes in their plastomes, except some housekeeping genes. A total of ten genome rearrangements were found among ten *Vanilloideae plastomes* when compared to the basal Apostasioideae plastomes. Both synonymous and nonsynonymous substitution rates of the IR in-cooperated SC sub-regions were decelerated, while the substitution rates of the SC in-cooperated IR sub-regions were accelerated. A total of 20 protein-coding genes commonly remained in mycoheterotrophic vanilloids. Almost all these protein genes show accelerated base substitution rates compared to photosynthetic vanilloids. Our data clearly supports the different evolutionary modes of plastomes between photosynthetic and nonphotosynthetic lineages.

## 5. Materials and Methods

### 5.1. Plant Material and DNA Extraction

The plant leaf materials used in this study and their voucher information are given in Table 1. Five plant samples were collected in nature, and one plant sample (*Pogonia japonica*) was obtained from artificial cultivation. All voucher specimens and DNAs were deposited into the KUS herbarium and the Plant DNA Bank in Korea. Fresh leaf materials were ground into a powder with liquid nitrogen in a mortar. The ground samples were used to extract genomic DNA, using a G-spin™II for Plant Genomic DNA extraction kit (iNtRON, Seoul, Korea). The quality of genomic DNA was checked with a UV/VIS spectrophotometer (Thermo, Wilmington, DE, USA).

### 5.2. Sequencing, Assembly, and Annotation

Two samples—*Vanilla madagascariensis* and *V*. *planifolia*—were sequenced by the Illumina HiSeq 2000 (Illumina, San Diego, CA, USA). The other four samples—*Lecanorchis kiusiana*, *L*. *japonica*, *Pogonia japonica*, and *P*. *minor*—were sequenced by the Illumina MiSeq (Illumina, San Diego, CA, USA). The raw reads from the Illumina HiSeq were trimmed by Geneious 6.1.8 [52] with an option of 0.05 for the error probability limitation. Trimmed reads were assembled by reference guided assembly using a Geneious assembler; *Vanilla planifolia* (NC026778) was used as a reference sequence. The raw reads from the Illumina MiSeq were trimmed by BBDuk 37.64 and imported in Geneious 11.1.5 (Kmer length of 27). The trimmed reads were normalized by BBNorm 37.64 (with a target coverage level of 30 and a minimum depth of 12). Normalized reads were used to perform the de novo assembly by the Geneious assembler. Both result contigs, which came from HiSeq and MiSeq, were used as reference sequences to filter the trimmed reads. Filtered reads were re-used to conduct the de novo assembly using a Geneious assembler to construct a complete plastome sequence. Obtained plastome sequences were compared with a previous result contig and assembled by reference guided assembly. Two *Lecanorchis* reads were reassembled to check the plastome structure using SPAdes 3.10.0 with error correction and the assembling method.

Complete plastome sequences were annotated with BLASTn, tRNAscan-SE 2.0 [53], ORF finder, and the find annotation function in Geneious 11.1.5 (NC026778—*Vanilla planifolia* was used as a reference). The alternative start codons, such as ACG and TTG, were also included in the ORF finder. In order to judge a protein-coding gene, pseudogene, and its absence, we used the criteria from a previous study [20]. Translated ORFs were extracted to perform psi-blast [54] with orchid plastome CDS. Circular plastome maps were constructed using the OGdraw web server [55].

### 5.3. Phylogenetic Analysis

In addition to our 6 vanilloid plastomes, 83 plastome sequences were downloaded from the NCBI to perform a phylogenetic analysis (Appendix A, 74 sequences for Orchidaceae, four species for Asparagales, one species for Liliales, and four species for Vanilloideae). The 79 CDS and four rRNA genes were extracted from the 89 plastomes. Extracted gene regions were aligned using MAFFT [56]. Furthermore, 83 independent gene alignments were concatenated to be a single length of 87,230 bp. The concatenated alignment was used to perform jModeltest2 [57] in the CIPRES science gateway [58] to obtain the best model. The best-fit model for concatenated data was the GTR + G + I model. A maximum likelihood (ML) tree was constructed using RaxML-HPC2 on XSEDE in the CIPRES science gateway [59] and 100 bootstrap replicates of ML optimization likelihood (−563,601.727423). The extracted 83 regions were aligned by MUSCLE [60] and used to perform PartitionFinder2 [61] to estimate the best-fit model by region. Eighty-three alignments were concatenated by the best-fit model (4.5S rRNA, 5S rRNA, *atp*H, *atp*I, and *psb*N for the GTR all-equal model; *clp*P, *pet*B, *pet*G, *pet*N, *psb*F, *psb*J, *psb*L, and *psb*T for the HKY estimated model; and the other 70 regions for the GTR estimated model). Concatenated alignments (83,665 bp length of the GTR estimated model, 1410 bp of the GTR all-equal model, and 2155 bp of the HKY estimated model) were used to construct a Bayesian inference phylogenetic tree by using MRBayes_CIPRES api [62,63] with 1,000,000 of chain length.

IRScope [64] was used to describe the SC-IR junction region among the Vanilloideae and four Apostasioideae species. All those fourteen species’ general features, such as the length of LSC/IR/SSC, were displayed on a graph by the ggplot2 package in R [65]. The progressiveMauve algorithm [66] was used to check gene relocation and plastome rearrangement among 14 species of Apostasioideae and Vanilloideae. The gene contents of 42 species in Orchidaceae were visualized as a heatmap by R.

Ten species of Vanilloideae and four species of Apostasioideae were used to estimate the synonymous and nonsynonymous substitution rates. Each region was aligned by MUSCLE [60]. The alignments were used to construct the maximum likelihood tree by the RaxML tree builder plugin in Geneious 11.1.5 with the GTR + G + I model. Alignments and phylogenetic trees were used to estimate the synonymous and nonsynonymous substitution rates by pamlX [67]. In order to estimate the position effect, regions were divided into two categories based on their movement from the original position to the IR region and from the original position to the SC region (LSC or SSC). A total of eight regions were counted as rearranged. Four regions—*rpl*2, *rps*7, *rps*12-3′, and *rps*19—moved to the SC region. Meanwhile, another four regions—*ccs*A, *rpl*22, *rps*15, and *ycf*1—moved to the IR region. One ratio omega model along all species was used as a null model. This alternative model adopts a different omega value between moved taxa (to the SC or IR regions) and consistent taxa (runmode = 0, model = 1, codon frequency = 3). The likelihood ratio test was performed to validate the statistical difference between the null model and the alternative model (Appendix A).

In order to estimate the mycoheterotrophic effect, the remaining 20 regions in all 14 species were used. Each sequence was aligned by MUSCLE, and the alignments were used to construct a maximum likelihood tree using the RaxML tree builder plugin with the GTR + G + I model. Alignments and phylogenetic trees were used to estimate the synonymous and nonsynonymous substitution rates by pamlX. One ratio omega model along all species was used as a null model. The alternative model adopts different omega values between mycoheterotrophic and photosynthetic taxa (runmode = 0, model = 1, codon frequency = 3). The likelihood ratio test was performed to validate the statistical difference between the null model and the alternative model (Appendix A).

A total of 28 regions (20 regions from the mycoheterotrophic constant region and eight regions for rearrangement) were used to perform RELAX in the Datamonkey server [68,69,70,71]. The selection intensity parameter and synonymous/nonsynonymous substitution rates between the reference branch (photosynthetic plants or constant regions) and test branch (mycoheterotrophic plants or rearranged regions) were checked.

### 5.4. Time Estimation

The alignments used in the Bayesian inference tree construction were used to estimate the divergence time estimation. Eighty-three regions were used (4.5S rRNA, 5S rRNA, *atp*H, *atp*I, and *psb*N for the GTR all-equal model; *clp*P, *pet*B, *pet*G, *pet*N, *psb*F, *psb*J, *psb*L, and *psb*T for the HKY estimated model; and the other 70 regions for the GTR estimated model). The XML file was prepared by BEAUti 2.5.2 with three fossil data to calibrate the nodes (Asparagales, normal distribution, mean 105.3, sigma 8.0; *Dendrobium*, log-normal distribution, sigma 2.0, offset 23.2; *Goodyera*, log-normal distribution, sigma 2.0, offset 15.0) [51,72,73]. A relaxed clock log-normal model [74] and a yule model were chosen to perform Markov chain Monte Carlo (MCMC) 100,000,000 times. Two independent analyses were performed by BEAST2-XSEDE in the CIPRES science gateway [75]. Trees and logs were collected for every 5000 generations.

The total log file was tested to check the effective sample size (ESS) by Tracer v1.6 [76], and the major parameters (posterior, prior, likelihood) were over 100 of the ESS. Tree files were concatenated by LogCombiner v2.5.2 [77] with an option of a 15% burn-in. A concatenated tree file was treated by TreeAnnotator [78] in the CIPRES science gateway with an option of 0.95 posterior probability. The concatenated maximum clade credibility tree was annotated by FigTree v1.4 [79] and two R packages of ‘phytools’ and ‘ape’ [80,81].

## Figures and Tables

**Figure 1 ijms-24-03808-f001:**
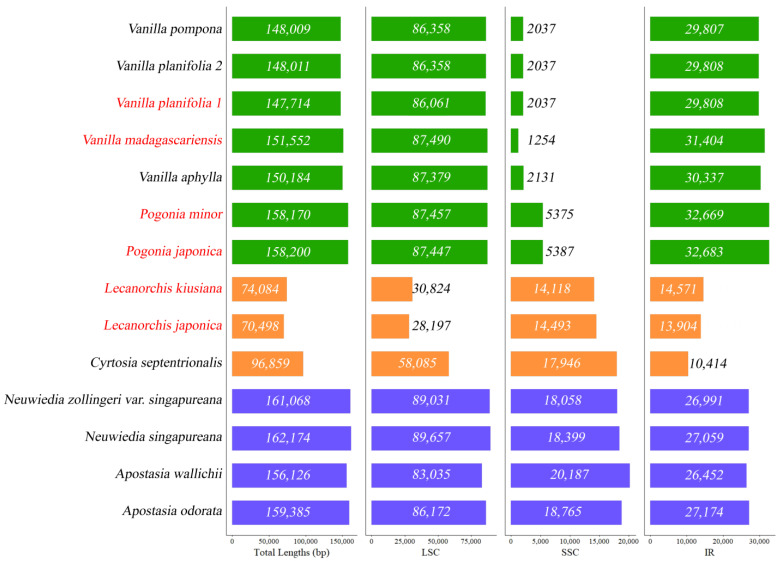
Plastome size variation among 14 species of Vanilloideae and Apostasioideae. The red scientific names indicate newly decoded plastomes. Green bars indicate the photosynthetic species, and orange bars indicate the mycoheterotrophic species. Blue bars indicate the four Apostasioideae species.

**Figure 2 ijms-24-03808-f002:**
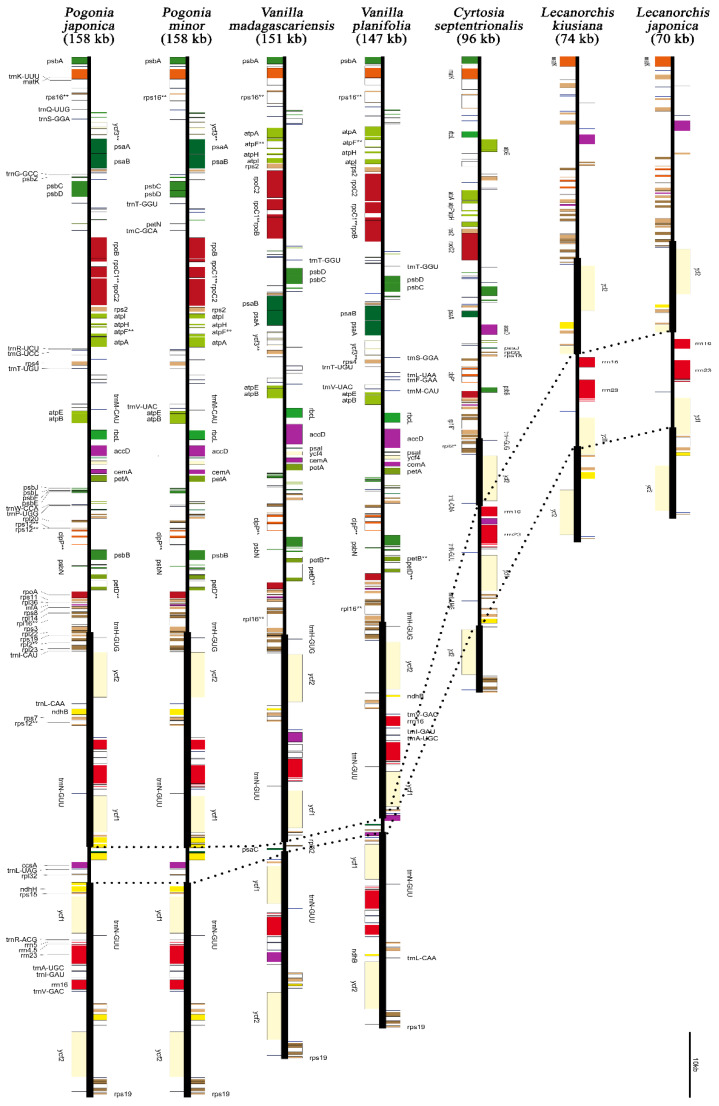
Seven representative plastomes of Vanilloideae. *Pogonia japonica* has the longest plastome, with 158.2 kb. *Pogonia* and *Vanilla* are the photosynthetic species, while *Cyrtosia* and *Lecanorchis* are the nonphotosynthetic mycoheterotrophic species. A few photosynthesis-related genes or pseudogenes remain in *Cyrtosia*, while all photosynthesis-related genes have been lost in *Lecanorchis*. Green gene boxes indicate genes related to the light reaction of photosynthesis. *L*. *kiusiana* and *L*. *japonica* hold only 33 and 32 gene regions, respectively. The heavy vertical lines indicate inverted repeat (IR) regions. The dashed lines indicate the single copy (SSC) regions.

**Figure 3 ijms-24-03808-f003:**
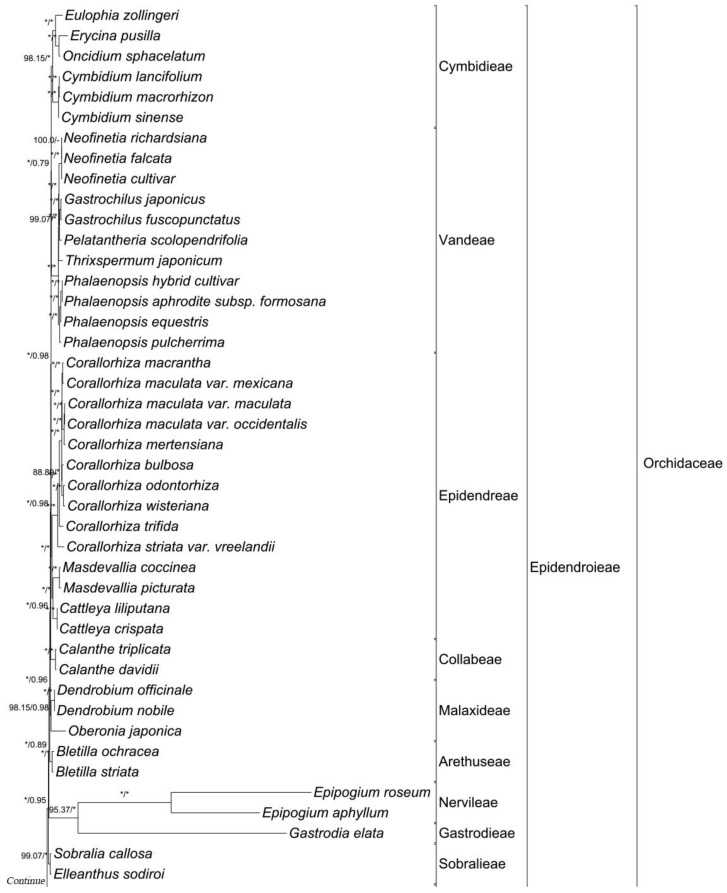
A maximum likelihood (ML) tree inferred from 89 species of Orchidaceae. The nucleotide sequences of 79 protein-coding genes and 4 rRNA genes were aligned individually and concatenated to be a single matrix. The aligned matrix was 87,230 bp, and the tree was reconstructed by RaxML with the GTR + G + I model (*−*563,601.727423 of ML value). The scientific names with red color indicate six newly decoded Vanilloideae species. This is an abbreviated tree, so the taxa of Cypripedioideae, Orchidoideae, and Epidendroideae are only given as a number of used species. The full tree structure is given in Appendix A. An asterisk (*) indicates that 100% of bootstrap value or 1 of Bayesian posterior probability.

**Figure 4 ijms-24-03808-f004:**
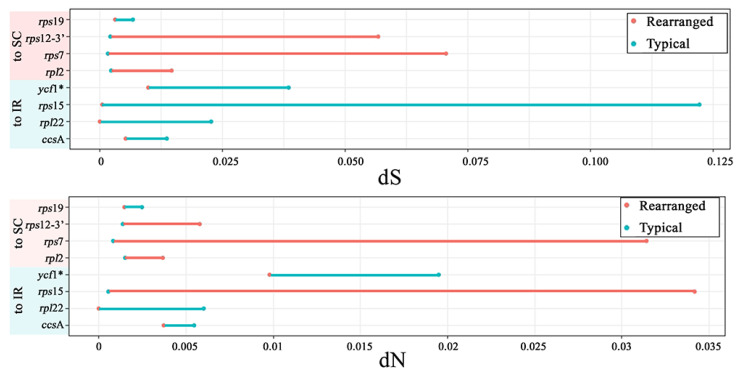
Comparison of synonymous (dS) and nonsynonymous (dN) substitution rates between species with rearranged and conserved plastomes in eight regions. A region with a significant *p*-value (*p*-value < 0.05) was tagged by an asterisk (*) in its gene name. The red color in the bar graph indicates an acceleration of the substitution rates; the blue color indicates a deceleration of the substitution rates.

**Figure 5 ijms-24-03808-f005:**
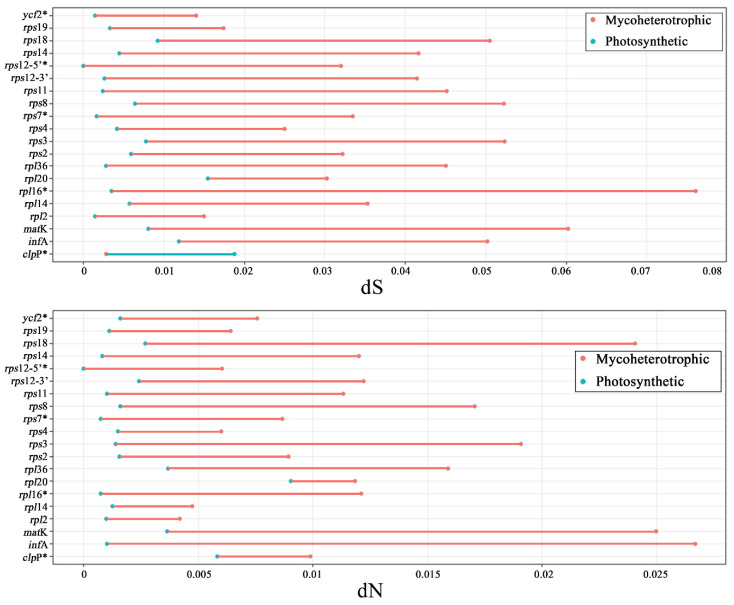
Comparison of synonymous/nonsynonymous substitution rates between mycoheterotrophic species and their photosynthetic relatives in the 20 commonly remaining protein-coding gene or gene regions. A region with a significant *p*-value (*p*-value < 0.05) is tagged by an asterisk (*). Red line indicates an acceleration of the substitution rates; blue line indicates a deceleration of the substitution rates.

**Figure 6 ijms-24-03808-f006:**
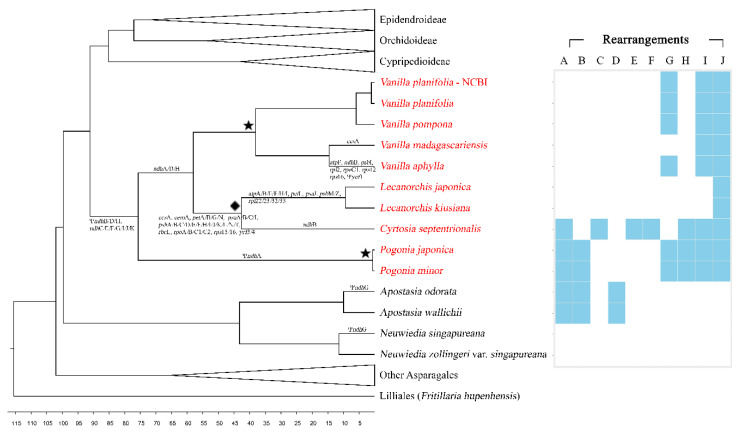
Summary of the rearrangements, mycoheterotrophy, SSC contraction, and gene reduction in the time estimation tree. A list of deleted or pseudogenized genes is above or beneath the tree nodes. A diamond mark in the phylogenetic tree indicates mycoheterotrophy of the clade. Star marks in the phylogenetic tree indicate the SSC contraction. Red scientific names indicate a Vanilloideae species.

**Table 1 ijms-24-03808-t001:** The general features of six newly decoded Vanilloideae species.

Scientific Name	Voucher Specimen and DNA Number	Total Length (bp)	Coverage (x)
*Lecanorchis japonica*	PDBK2018-0250	70,498	76.9
*Lecanorchis kiusiana*	PDBK2018-0249	74,084	128.9
*Pogonia japonica*	PDBK2015-1272	158,200	464.6
*Pogonia minor*	PDBK2011-1673	158,170	563.2
*Vanilla madagascariensis*	PDBKTMA2013-1861	151,552	4394
*Vanilla planifolia*	PDBKTMA2013-1860	147,714	592

**Table 2 ijms-24-03808-t002:** Rearrangement blocks in the result of progressiveMauve among fourteen species (ten Vanilloideae species and four Apostasioideae species were used).

Rearrangement	Length	Region
A	3.282	*ycf*3-*trn*S(GGA)
B	58.591	*trn*S(GCU)-*ycf*3
C	24.647	*trn*S(GCU)-*rbc*L
D	6.658	*acc*D
E	7.199	*ndh*B-*rps*12
F	4.037	*trn*V(GAC)
G	4.626	*rpl*32-*ndh*F
H	8.899	*trn*L(UAG)-*ndh*A
I	6.469	*ycf*1-*ndh*A
J	7.284	*ycf*1

**Table 3 ijms-24-03808-t003:** Estimated selection pressure in the result of RELAX. Regions with significant p-values are listed in the table.

Type	Gene	Selection	k	*p*-Value	LR	w1	w2	w3
Myco hetero trophs	*clp*P	Relaxation	0.39	0.022	5.27	0/0 (78.27%)	1/1 (4.27%)	8.02/2.25 (17.47%)
	*ycf*2	Relaxation	0.45	0	49.96	0.95/0.98 (97.53%)	164.08/9.95 (2.47%)	-
Regional	*rpl*2	Intensification	4.22	0.015	5.97	0.44/0.03 (20.13%)	0.58/0.10 (58.93%)	1.00/1.46 (20.95%)
	*rps*15	Relaxation	0	0.003	8.74	0.27 (20.26%)/1.00 (2.64%)	0.30 (77.09%)/1.00 (20.26%)	32.11 (2.64%)/1.00 (77.09%)
	*rps*12-3’	Relaxation	0	0.048	3.91	0.00/0.00 (80.01%)	0.55/1.00 (15.17%)	292.30/1.00 (4.82%)
	*ycf*1	Relaxation	0.85	0.022	5.24	0.00/0.01 (24.94%)	1.00/1.00 (72.32%)	192.01/86.05 (2.74%)

## Data Availability

The six newly sequenced Vanilloideae species (*Lecanorchis kiusiana*, *L*. *japonica*, *Pogonia japonica*, *P*. *minor*, *Vanilla madagascariensis*, and *V*. *planifolia*) were submitted to GenBank with accession numbers of MN200375, MN200374, MN200371, MN200372, MN200374, and MN200375, respectively.

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
