# Peer review of "Evolutionary Patterns of the Chloroplast Genome in Vanilloid Orchids (Vanilloideae, Orchidaceae)"

_ijms, 2023, doi:10.3390/ijms24043808_

Round 1

Reviewer 1 Report

Kim et al. used plastomes to reconstruct the phylogeny of vanilloids, and analyis the divergence time, evolution rate and selection pressure. They found that vanilloids’ plastome contracted in SSC and plastome degradation occurs in both photosynthetic and mycoheterotrophic species, and plastomes rearranged from typical orchis. The mycoheterotrophic species were at higher degradation level and had lost most genes except the housekeeping genes. I have some questions.

1.      The plastome feature of Vanilloideae are different from the other Orchidaceae taxa, especially the SSC contraction event. This result indicated that Vanilloideae is a particular group in Orchidaceae. But the authors compared the subfamily with only Orchidaceae species. I wonder if there are any similar features in other taxa, and cases in other heterotrophic angiosperms might provide some trails of possible mechanism.

2.      Line 513. More detail information of the three fossil data should be present.

Minors

1.      Line 285. “Orchidaceae” might be “Vanilloideae”.

2.      Line 362. “mycohetertrophic” should be “mycoheterotrophic”.

3.      Line 415. “Aphyllorchis (Epidendreae)” should be “Aphyllorchis (Neottieae)”.

4.      Line 460. “The 78 CDS and four rRNA genes were extracted from the 89 plastomes” The total number genes are 82 not 83.

Author Response

A. The plastome feature of Vanilloideae are different from the other Orchidaceae taxa, especially the SSC contraction event. This result indicated that Vanilloideae is a particular group in Orchidaceae. But the authors compared the subfamily with only Orchidaceae species. I wonder if there are any similar features in other taxa, and cases in other heterotrophic angiosperms might provide some trails of possible mechanism. 

>>> Two new sentences were added in line 279 and 280.

B. Line 513. More detail information of the three fossil data should be present.

>>> We provide three references for fossil data.

C. Line 285. “Orchidaceae” might be “Vanilloideae”.

>>> Yes. Changed.

D. Line 362. “mycohetertrophic” should be “mycoheterotrophic”.

>>> Yes. The typo was corrected.

E. Line 415. “Aphyllorchis (Epidendreae)” should be “Aphyllorchis (Neottieae)”.

>>> Yes. Changed.

F. Line 460. “The 78 CDS and four rRNA genes were extracted from the 89 plastomes” The total number genes are 82 not 83.

>>> We combined 79 CDS and four rRNA genes. Therefore, 78 was corrected as 79 instead of the 83.

Reviewer 2 Report

Comments on IJMS (ISSN 1422-0067) entitled “Evolutionary patterns of the chloroplast genome in 2 vanilloid orchids (Vanilloideae, Orchidaceae)”

The study is on the evolutionary aspects of six vanilloids (two Lecanorchis, two Pogonia, and two Vanilla species) with emphasis placed on their plastome evolution pattern.

The study is interesting as it throws light into the  Plastomes which have a  variety of functions, including photosynthesis, storage of starch and pigments, and the synthesis of amino acids and lipids. The evolutionary pattern is all the more important as the plastomes  encodes a variety of genes, including those involved in photosynthesis, the biosynthesis of pigments, and the regulation of plastid development.

It provides  clear overview of the study's objectives and methods. It is well-organized, with each point of the study's goals being described in a concise and clear manner. The use of specific examples of the new plastomes documented and the modifications they exhibit is helpful in giving a sense of the scope and focus of the study. The inclusion of a phylogenetic analysis and a comparison of synonymous and non-synonymous substitution rates adds to the study's rigor and provides a more complete picture of the evolution of vanilloid plastomes. However, the  author has to be a little more specific about the research question and the hypothesis that the study is trying to test.

The results section is well written with all the details of the results obtained in a clear manner

The discussion section is well written. This discussion on  the results that compares substitution rates in various regions of the plastome between reference and test groups of angiosperms is clear. The authors bring out the relationships in regions that were moved to the Inverted Repeat (IR) showing lower synonymous substitution rates, while regions moved to the Single Copy (SC) show higher rates, is written well. The suggestion  that selection pressure may be affecting the elevated substitution rates, and that two regions show significant p-values under relaxation selection pressure, and four regions under position effect also showed significant p-values, with different selection pressures is brought out by the authors.

In page 16, lines 394 – 395 the authors say that “Vanilloideae contains photosynthetic and  mycoheterotrophic species, its branch length is expected to be normal” a sentence on how the  photosynthetic and  mycoheterotrophic species  have normal branch length can be given.

Materials and Methods section is well written and the details of the materials used is given in detail. The experiment is well planned and executed.

The study has come out with some novel findings.

The authors can add a paragraph of conclusion at the end.

The MS can be accepted with minor revisions.

Author Response

A. However, the author has to be a little more specific about the research question and the hypothesis that the study is trying to test.

>>> In order to describe the research goals more specifically, we modified the last paragraph of introduction.

B. In page 16, lines 394 – 395 the authors say that “Vanilloideae contains photosynthetic and mycoheterotrophic species, its branch length is expected to be normal” a sentence on how the photosynthetic and mycoheterotrophic species have normal branch length can be given.

>>> To clarify the discussion, the sentences between lines 402-405 was modified.

C. The authors can add a paragraph of conclusion at the end.

>>> We add a conclusion section in the manuscript.

Reviewer 3 Report

The manuscript reports the sequence of chloroplast genomes of six species of the Vanilloideae subfamily of Orchidaceae and compare with previously reported sequences of orchids, especially other vanilloids. The manuscript is easily readable, and the experimental approach and analyses of data are conventional and straightforward as follow from the use of available kits and application software.

Focused on vanilloid subfamily, the manuscript does not report special novelties when compared with the many articles published during the last ten years on chloroplast DNA sequences of Orchidaceae, their comparative analyses, phylogenetic and adaptative implications. Re-arrangements, gene loss and pseudogenization have being yet published in many Orchidaceae chloroplast DNA. Embarked in an evolutionary transition from autotrophy to heterotrophy, many Orchidaceae are dismantling the chloroplast genome, as is evident in mycoheterotrophic vanilloids. The adaptative advantages of gene loss is low and, therefore, they occur mainly randomly, which provide a valuable taxonomic tool. However, gene loss was not completely hazardous, constrains of the chloroplast genome structure and functional roles of specific groups of genes determined some order of the successive gene loss whose identification provide clues of past evolutionary steps and explanation to present functional adaptations. With this aim, authors should compare their results with previous publications that must be cited.

The manuscript should go beyond the taxonomic perspective of the results and discuss phylogenetic and adaptative explanations and their connection to different lifeforms. This approach would improve the low “Biochemistry” content in one manuscript expected for the “Special Issue: Orchid Biochemistry”.

Minor points.

Lines 107-109. In the sentence: “Almost all the genes except for the ndh genes of four photosynthetic species (Pogonia japonica, P. minor, Vanilla madagascariensis, and V. planifolia) were found to be pseudogenized or lost. In contrast …” . The words “except” and “In contrast” are confusing and do not describe facts. Please, re-write.

Line 132-133. Delete “horizontal”. Substitute “lidicate” by “indicate”.

Author Response

A. With this aim, authors should compare their results with previous publications that must be cited.

>>> We add the references citations [10, 11, 12, 13, 15, 23, 24, 25, 26, 27, 28, 40, 41, 42, 50] to the corresponding paragraphs of the manuscript.

B. The manuscript should go beyond the taxonomic perspective of the results and discuss phylogenetic and adaptative explanations and their connection to different lifeforms. This approach would improve the low “Biochemistry” content in one manuscript expected for the “Special Issue: Orchid Biochemistry”.

>>> It is a very good point. We modified the research goals, expended or clarify the discussion, and finally add a conclusion section.

C. Lines 107-109. In the sentence: “Almost all the genes except for the ndh genes of four photosynthetic species (Pogonia japonicaP. minorVanilla madagascariensis, and V. planifolia) were found to be pseudogenized or lost. In contrast …” . The words “except” and “In contrast” are confusing and do not describe facts. Please, re-write.

>>> We modified the sentences on lines 108-109 to clear description.

D. Line 132-133. Delete “horizontal”. Substitute “lidicate” by “indicate”.

>>> Yes. Corrected.